# Caregiver Burden in Small Animal Clinics: A Comparative Analysis of Dermatological and Oncological Cases

**DOI:** 10.3390/ani14020276

**Published:** 2024-01-16

**Authors:** Pollyana T. R. F. Silva, Fernanda M. Coura, Adriane P. Costa-Val

**Affiliations:** 1Escola de Veterinária, Departamento de Clínica e Cirurgia Veterinária, Universidade Federal de Minas Gerais, UFMG, Av. Antônio Carlos, 6627, Belo Horizonte 30123-970, Brazil; pollyana-torres@ufmg.br; 2Instituto Federal de Educação, Ciência e Tecnologia de Minas Gerais Campus Bambuí, IFMG, Faz. Varginha, Rodovia Bambuí/Medeiros, Km 05, Caixa-Postal 05, Zona Rural, Bambuí 38900-000, Brazil; fernanda.coura@ifmg.edu.br

**Keywords:** burden of care, cat, dog, human–animal interaction, owner

## Abstract

**Simple Summary:**

This study investigated the challenges faced by caregivers of sick small animals, exploring the burden experienced by those caring for pets with dermatological and oncological issues. Through surveys conducted at a veterinary hospital, this study revealed that a significant portion of caregivers, irrespective of their pets’ conditions, faced substantial burdens, impacting their emotional, physical, and financial well-being. This research found that factors like disease stability, time since diagnosis, and guardian age played a crucial role in determining the extent of caregiver burden. However, the type of pet (cat or dog), income, and treatment duration had less influence on the experienced burden. These insights shed light on the often-overlooked challenges faced by pet caregivers, providing valuable information for veterinarians to enhance communication and tailor treatments, ultimately ensuring the well-being of both pets and their devoted caretakers.

**Abstract:**

Caregiver burden, a response to the challenges faced by those caring for sick loved ones, combines objective caregiving aspects with subjective experiences. This study aimed to describe the caregiver burden in guardians of ailing animals with dermatological and oncological pathologies. Additionally, this study aimed to correlate the degree of caregiver burden with the duration of the animal’s treatment, disease stability, family income, owner’s age, and the number of people living in the same household responsible for the animal’s treatment. Using a cross-sectional approach, questionnaires were administered to guardians at a veterinary hospital. Out of 182 valid responses, 50.55% related to oncological patients and 49.45% to dermatological patients. Notably, 36.9% of oncological and 37.8% of dermatological guardians exhibited a clinically significant burden, with no statistical differences between the groups. Of the respondents, 84.6% were dog guardians, with 34.4% showing a significant burden. Feline guardians (15.4% of respondents) exhibited a higher burden (53.5%) compared to dog guardians (*p* = 0.02). Disease stability, time since diagnosis, and guardian age significantly influenced the burden, while household composition, income, and treatment duration showed no substantial impact. These findings underscore the psychosocial impact of caring for animals, emphasizing the need for a comprehensive understanding of the caregiver’s perspective.

## 1. Introduction

The caregiver burden is a response to the numerous problems and challenges faced by individuals caring for a sick loved one [1,2]. This burden is considered a combination of both the objective aspects of caregiving, including the time and physical demands of care, as well as subjective experiences such as emotional perceptions secondary to individual care [3,4].

This phenomenon was initially described in 1980 by Zarit and colleagues in the medical field, particularly when associated with chronic and terminal illnesses such as cancer, Alzheimer’s disease, and stroke [1]. These researchers developed a 22-question questionnaire to assess this burden, known as the Zarit Burden Interview (ZBI).

In 2017, Spitznagel and colleagues evaluated this burden and the psychosocial function of pet owners of sick animals through a modified ZBI for pet caregivers—an 18-item questionnaire graded on a five-point scale ranging from 0 (never) to 4 (almost always). A total score above 18 on the modified ZBI is considered indicative of a clinically significant burden. The results showed increased burden, stress, and symptoms of depression/anxiety, as well as a poorer quality of life in caregivers of animals with a chronic or terminal illness. The higher burden was correlated with a reduced psychosocial function, i.e., emotional and social impairment of these caregivers [4].

The findings from Belshaw et al. [5] and Spitznagel et al. [6] shed light on the caregiver burden experienced by owners of dogs diagnosed with osteoarthritis. Belshaw et al. revealed that owners faced heightened worry and concern due to factors such as veterinary visits, challenges in finding effective treatments, and managing their dog’s medication and weight. Negative consequences, including physical health issues like back pain, were reported. Despite these challenges, owners expressed enduring love and devotion, emphasizing the emotional support provided by their dogs in times of stress and life changes. Spitznagel et al. suggested that owner satisfaction with treatment and the veterinary healthcare team played a protective role against caregiver burden, influencing decisions related to euthanasia. Satisfactory treatment and support offset considerations of euthanasia, emphasizing the importance of addressing owner satisfaction and providing comprehensive support to mitigate caregiver burden and its potential impact on decision-making processes.

The caregiver burden experienced by owners of dogs with behavioral problems was also addressed. Buller and Ballantyne [7] indicated consequences such as an increased time for management and training, limitations on activities, and emotional challenges like sadness, frustration, and guilt. Despite a strong human–animal bond, these negative emotions could lead to social isolation. Kuntz et al. [8] extended this understanding by demonstrating that 68.5% of owners experienced a clinically meaningful caregiver burden. Older owners reported less burden, indicating age-related variations. This study also highlighted the potential burden transfer to veterinarians in behavior practices, emphasizing the need for routine evaluation to address occupational distress. Both studies underscore the complex emotional landscape of owners caring for dogs with behavioral issues, emphasizing the importance of tailored support and interventions for their well-being.

Spitznagel et al. [9,10] analyzed the caregiver burden experienced by owners of dogs with chronic or terminal diseases and identified signs and behaviors correlated with burden, including weakness, the appearance of sadness or anxiety, pain or discomfort, and personality changes. Caregiver burden was associated with greater symptoms of depression and stress, indicating a significant impact on psychosocial function. Also, changes in routines due to the pet’s condition and the perceived difficulty of following new rules and routines for management were linked to increased caregiver burden. Recognizing caregiver burden and understanding its determinants can inform empathic responses from veterinarians, fostering better communication and potentially alleviating downstream effects, such as an increased workload for veterinary professionals. In a cross-sectional observational study involving 164 owners of cats or dogs undergoing evaluation at a veterinary oncology service, the researchers found a significant correlation between caregiver burden and elevated stress, increased symptoms of depression, and a lower overall quality of life. Additionally, the highest level of burden was noted when associated with certain treatment plan factors, such as changes in animal care routines, with the perception that the adherence to treatment routines was challenging, and/or with a feeling of difficulty in adhering to medication routines. The authors emphasize the importance of recognizing caregiver burden in individuals caring for companion animals with cancer, even in its early stages [11].

Dermatological diseases in pets often require long-term treatment that may contribute to caregiver burden. Considering this, Spitznagel and colleagues [12] in 2017 assessed the caregiver burden in pet owners of animals with dermatological conditions compared to owners of healthy animals and the relationship of these caregivers with quality of life. They found a correlation between caregiver burden and quality of life related to dermatological disease, indicating a higher burden in caregivers of animals with skin diseases compared to healthy animals [12]. The findings from the 2021 study [13] revealed a significant correlation between caregiver burden and both skin disease severity and the complexity of the treatment plan, emphasizing the importance of simplicity in treatment planning. Even after considering skin disease severity, the correlation with treatment complexity remained significant, suggesting that starting with the simplest effective treatment may reduce owner strain and improve long-term management for the dog. The 2022 study [14] further explored the connections among caregiver burden, treatment complexity, and the veterinarian–client relationship. It supported the idea that a greater treatment complexity is related to the owner’s perception of the veterinarian–client relationship through caregiver burden. The indirect relationship observed between treatment complexity and the veterinarian–client relationship implies that reducing the burden of complex treatment may foster a better working relationship between the veterinarian and client, enhancing perceptions of care, compassion, and trustworthiness.

Studies on caregiver burden in cats are relatively scarce compared to those focusing on dogs. A recent study [15] added valuable insights by examining the caregiver burden of cat owners. This study hypothesized that owners of an ill cat would experience a greater caregiver burden compared to owners of a healthy cat but a lower burden than owners of an ill dog. The results supported these hypotheses, revealing that owners of an ill cat, across various illnesses, indeed exhibited a greater burden than owners of a healthy cat and a somewhat lower burden than owners of an ill dog. The reasons behind this disparity are not fully clear, but the authors suggest that a potential lack of veterinary oversight in cat care may contribute to fewer caretaking requirements for cat owners. Importantly, while this difference is statistically significant, it might not have a clinically meaningful impact on the interpretation. Henning et al. [16] investigated factors associated with the quality of life (QOL) in cats with epilepsy and the burden of care in their owners. This study highlighted the crucial role of veterinarian support, emphasizing that owners who felt unsupported might be reluctant to seek veterinary care, resulting in uncontrolled seizures and increased caregiver burden. Additionally, the burden was reported to be higher in caregivers aged 18 to 34, possibly due to life-stage pressures, while older caregivers who had already established a career and settled were better equipped to provide care and experience fewer burdens.

Suspecting that the severity of the animal’s disease may influence the occurrence of caregiver burden, it was hypothesized that the degree of burden in caregivers of animals with oncological conditions is higher than that of caregivers of animals with dermatological conditions. Therefore, the objective was to compare the burden among caregivers of patients in these two specialties.

## 2. Materials and Methods

This research was approved by the Research Ethics Committee (CEP) under the Certificate of Presentation for Ethical Appreciation (CAAE) 58860122.0.0000.5149. This research was conducted through a cross-sectional study using questionnaires administered to owners of dogs and cats undergoing dermatological and oncological treatment at the institution in a veterinary hospital.

Inclusion criteria for study participation were individuals aged 18 years or older, capable of understanding and responding to the questionnaire, responsible for an animal undergoing treatment prescribed by a specialist, and present throughout the duration of the pet’s treatment. Participants who did not meet these criteria, did not sign the Informed Consent Form (ICF), or did not complete the questionnaire in full were excluded from this study. Participation in this research was voluntary.

Demographic information about the owner was collected and included age, gender, educational level, income, the number of people living in the household, and whether they were the sole caregiver for the animal. Information about the animal, such as species (cat or dog), diagnosis, time since diagnosis, disease stabilization, and treatment duration, was also collected through a questionnaire (Table 1).

The modified Zarit Burden Interview (ZBI) for pet caregivers was translated and used to assess caregiver burden [4] (Table 2). A total score above 18 on the modified ZBI was considered indicative of a clinically significant burden. However, to correlate the degree of overload in tutors of oncology and dermatology patients with other variables such as time of diagnosis or treatment, perception of disease stability, and the number of people living with the tutor and assisting in the treatment, additional novel questions were included in the tutor assessment. The questionnaires were administered in person after the animal’s consultation. Individuals who chose to participate in this study signed the ICF beforehand. After data collection, the information was compiled into tables for statistical analysis.

The sample size was calculated based on the formula for calculating sample sizes for the description of qualitative variables in a population [17]. Thus, a minimum sample of 67 individuals in each group was obtained. Due to the diverse probability distributions of the measured variables and the presence of outliers, non-parametric methods were applied to check for differences between groups. The Mann–Whitney test was applied to compare two groups, and non-parametric Spearman correlations were calculated between specific pairs of variables. All statistical analyses were performed using R software version 4.1.3 [18].

## 3. Results

### 3.1. Demographic

A total of 193 participants completed the survey. After excluding incomplete questionnaires, 182 valid responses were obtained—50.55% (92/182) related to oncological patients and 49.45% (90/182) to dermatological patients. Concerning the species of the patients, 84.6% (154/182) were dogs and 15.4% (28/182) were cats.

As for the caregivers, 72% (131/182) identified as female, 26.9% (49/182) identified as male, 0.5% (1/182) considered themselves non-binary, and 0.5% (1/182) preferred not to disclose. The majority of caregivers (67%, 122/182) reported that they did not consider their animal’s disease to be stable at that time. A significant portion of caregivers (51.6%, 94/182) were the sole individuals responsible for their animal’s treatment at home; however, this factor did not show statistical significance (*p* = 0.45224). In terms of the number of people residing in the same household, 11.6% (21/182) lived alone, 44.2% (80/182) lived with 1 to 2 people, 40.3% (73/182) lived with 3 to 4 people, and 3.9% (7/182) lived with more than 4 people.

Regarding education, 0.5% (1/182) of caregivers had an incomplete primary education, 0.5% (1/182) completed primary education, 1.1% (2/182) had an incomplete secondary education, 13.2% (24/182) completed secondary education, 16.5% (30/182) had an incomplete higher education, 30.2% (55/182) completed higher education, and 37.9% (69/182) had a postgraduate education. Regarding family income, 1.2% (2/182) had no family income, 0.6% (1/182) had an income up to two minimum wages, 42.2% (70/182) had 2 to 4 salaries, 34.9% had 4 to 10 salaries (58/182), 13.3% had 10 to 20 salaries (22/182), and 7.6% (13/182) preferred not to disclose.

### 3.2. Caregiver Burden

The degree of caregiver burden showed statistical differences for caregivers who reported that they did not consider their animal’s disease to be stable at that time compared to those who considered their pet’s disease to be stable (*p* = 0.00026). Importantly, 36.9% (34/92) of caregivers responsible for animals undergoing oncological treatment showed indications of a clinically significant burden. Regarding caregivers of dermatological patients, 37.8% (34/90) scored above 18 on the Zarit Burden Interview (ZBI) scale. There was no statistical difference in the degree of overload between the two specialties, although over a third of the tutors exhibited a significant degree of overload (score > 18) (Figure 1).

Regarding caregiver burden, 34.4% (53/154) of dog owners and 53.5% (15/28) of cat owners exhibited a significant level of burden. Statistically significant differences in the degree of burden were observed between the two groups (*p* = 0.01044). The correlations of caregiver burden with age, the number of people living in the same house, family income, and overall diagnosis and treatment duration are shown in Figure 2. In the overall assessment, there was statistical significance and a negative correlation of overload with age. Regarding the number of people living in the same house and the duration of treatment, no statistical significance was observed, and there was a weak positive correlation. As for family income, no statistical significance was observed, and there was a weak negative correlation. Finally, concerning the time of diagnosis, there was statistical significance and a positive correlation with the degree of overload.

Concerning the interplay between caregiver burden and key factors—age, household composition, family income, diagnosis duration, and treatment duration—according to the dermatological and oncological conditions, the results are shown in Figure 3. Due to the diversity of correlation values (R), these data should not be used for comparison between specialties. There was statistical significance when correlating overload with the age of tutors of oncology patients and with the time of diagnosis in dermatology patients. No statistical difference was observed in any of the specialties in the variables of the number of people living in the same house, family income, and treatment duration.

## 4. Discussion

Caring for a beloved pet with an illness or when approaching the end of their life can be challenging for many pet owners, especially considering the multifaceted nature of having a pet as a family member. While pets undoubtedly bring companionship and support, the increased needs resulting from illness or injury can amplify the inherent stresses associated with pet ownership [19]. This highlights the importance of exploring the caregiver burden among pet owners. To our knowledge, this study represents the first exploration of caregiver burden in companion animal owners in Brazil. This unique contribution adds valuable insights to the global understanding of the complex dynamics involved in caring for ill pets.

The present study demonstrated that over a third of the caregivers of animals undergoing treatment for oncological and dermatological pathologies experienced a significant level of burden at the time of evaluation, although no statistical differences were found in the degree of burden between these two veterinary specialties. In both cases, dealing with chronic conditions requires prolonged and often emotionally, physically, and financially demanding treatments, such as multimodal therapies for dermatological cases and chemotherapy for oncological patients. Caregivers may feel overwhelmed by recommended treatments, even when deeply attached to their pets, emphasizing the need for understanding and effective communication in the treatment process [9].

Similar to the scenario of human patient caregivers, women are typically the primary caregivers for pets [20]. A study conducted by the American Veterinary Medical Association (AVMA) in 2012 showed that 74.5% of caregivers primarily responsible for these animals were female, a trend also observed in the current study [21].

Spitznagel et al. [12] found a higher degree of burden in caregivers of animals with dermatological pathologies compared to caregivers of healthy animals. Additionally, caregivers who perceived their animal’s disease as controlled or stable had a burden level (not elevated/normal) statistically similar to that of healthy patients. Disease stability was also a statistically significant factor related to burden in this study, emphasizing the importance of proper treatment in this population.

It was expected that sharing the responsibility for animal treatment with another individual would result in a lower degree of burden; however, there was no statistical difference between the degree of burden and whether the caregiver was solely responsible for the animal’s treatment in the current study. Similarly, there was also no difference in the number of people sharing the same household. A possible explanation for this phenomenon could be the uneven distribution of care responsibilities for these patients, resulting in a higher burden for the caregivers who responded to the questionnaire.

Another potential explanation for the no statistical difference in caregiver burden based on shared responsibilities might be rooted in the uneven distribution of care responsibilities for these patients. This study calculated the size of the experiment, considering that those who responded to the questionnaire are likely to be individuals more predisposed to experiencing caregiver burden. This inclination could be attributed to their role as the primary person responsible for tasks such as driving the animal to the vet, administering treatments, and managing the day-to-day care, thereby potentially intensifying their perceived burden in comparison to those who may share these responsibilities. This aspect underscores the importance of recognizing the diverse roles within caregiving and suggests that further exploration of these dynamics is crucial for a comprehensive understanding of caregiver burden in the context of pet ownership and treatment.

Despite no statistical difference being observed in the degree of burden with treatment duration (overall and in both specialties), there was a trend of increased burden with the animal’s therapy duration (positive correlation). Shaevitz et al. [11] conducted a study assessing caregiver burden in caregivers of animals suspected of having cancer. It was observed that, in these cases, burden is present in the early stages of the disease. Furthermore, the results suggested that caregiver burden is similar in pet owners with cancer and pet owners with other diseases.

A statistical difference was observed in both the overall context and specifically among animals with dermatological diseases concerning the duration of diagnosis and the associated caregiver burden. This disparity was not evident in caregivers tending to animals with neoplasms. An explanation for this occurrence can be found by examining the difference in the diagnosis duration between these two specialties. The average diagnosis duration of animals with dermatological disorders (maximum of 144 months) was almost double that of animals with oncological pathologies (maximum of 48 months), which may be correlated with the difference in the diagnostic process between these two specialties. The association between treatment plan factors, such as changes in routines and the perceived challenge of following new rules for managing the animal’s condition, underscores the significance of treatment planning in influencing caregiver burden and highlights the importance of tailoring treatment strategies to align with the caregiver’s lifestyle and capabilities, thereby minimizing the potential negative impact on their caregiving experience [9].

In most cases, animals with oncological diseases, once diagnosed, are referred to specialists to assess and/or initiate appropriate treatment. Regarding dermatological pathologies, many patients arrive at the specialist after numerous attempts at treatment with the general practitioner, and often the caregiver believes they already have an adequate diagnosis when, in many cases, what the animal presents is secondary to the main disease yet to be diagnosed, as in the case of otitis secondary to allergies [22], where the caregiver believes that otitis is the final diagnosis in itself. Thus, caregivers often spend years treating a recurring problem without a proper resolution, implying a longer “diagnosis” time and burden.

In the current study, no statistical difference was found in any group between the degree of burden and the caregiver’s income. However, there was a negative correlation between the variables, meaning the higher the income, the lower the burden. The surveyed sample reported a relatively high level of income and education (34.9% for 4 to 10 minimum wages and 37.9% postgraduate), reflecting the typical clientele of the professional who is a specialist: individuals who choose to take their animal to a referring veterinarian and who can prioritize the healthcare of their companion animals. Britton et al. [23] found that burden is significantly related to financial strain. This is because financial commitment is an important factor in this issue, as it can be both a risk for the development of the burden and a result of this burden [2,24].

Regarding the age of caregivers, a negative correlation was also found in all groups and was statistically significant in caregivers of animals with oncological pathologies and overall, which may reflect the trend of greater maturity and the ability to deal with adversities acquired over time. In a study with owners of dogs with behavioral problems, caregiver burden was significantly less reported in older owners [8]. In a study of caregiver burden in owners of dogs with cognitive dysfunction, the results demonstrated that those who lived alone and were between the ages of 25 and 44 years had an increased burden of care [25]. Henning et al. [16] found that the burden of care was lower in cat owners over 55 years of age and suggested that younger pet owners may experience a higher burden in caring for their animals compared to older caregivers due to specific life stage pressures for the younger caregiver, such as building a career, socializing, and childbearing; in contrast, older caregivers may find it easier to allocate time at home for pet care.

In the current study, a statistical difference was observed between the degree of burden in cat caregivers (higher burden) compared to dog caregivers. Spitznagel et al. [15] researched the occurrence of caregiver burden specifically in cat caregivers. It was observed that owners of sick cats had a higher burden than owners of healthy cats, but a slightly lower burden than owners of sick dogs, results that differ from ours. It was pointed out that the differences in the challenges of caring for these two species, such as giving baths—for example, dogs need more structure—can make it more difficult to care for the dog. Moreover, many cats do not receive routine veterinary care, even for chronic diseases, and this lack of veterinary oversight can lead to a lower demand for care for cat caregivers, but more studies to understand these differences are necessary.

The questionnaire of this study was administered to caregivers with animals at different stages of treatment—whether the diagnosis was recent or old. This was even an item evaluated to determine the correlation with the occurrence of burden (time of diagnosis and treatment). An interesting future study would be to assess the evolution of the burden level as treatment progresses, with repeated assessments of the same caregiver. Despite the average of questionnaire scores of approximately 16 points in both specialties being below what is considered a significant burden level, about one third of caregivers of animals with dermatological and oncological pathologies experienced a significant burden. Therefore, the importance of research in this area and the development of strategies to ease caregivers’ daily lives become evident. Additionally, treatments should be conducted in an empathetic and less traumatizing manner, adapted to generate less burden for these caregivers who experience delicate situations.

Examining caregiver burden in the context of veterinary patients is crucial for comprehending the roles and responsibilities of both clients and veterinarians, particularly in the treatment of seriously and terminally ill pets. The emotional labor in veterinary medicine, especially in client interactions surrounding serious or terminal illnesses, is substantial [26]. The provided studies predominantly concentrated on the experiences of dog owners, highlighting the need for further research dedicated to understanding the nuanced aspects of caregiver burden in feline companionship, potentially uncovering unique challenges and dynamics associated with caring for cats with various health conditions.

## 5. Conclusions

This study, the first of its kind in Brazil, examined the caregiver burden among companion animal owners. However, a significant portion of caregivers experienced a clinically significant level of burden. Therefore, regardless of the specific illness the patient has, caregivers are intensely impacted when dedicating themselves to the prolonged care of their pet. Numerous efforts are invested in hoping for the recovery of a loved one, which can contribute to financial, physical, and/or psychosocial exhaustion.

The findings unveil that over a third of caregivers for animals undergoing treatment for oncological and dermatological pathologies experience a significant burden. While no statistical differences were found between these veterinary specialties, the challenges associated with chronic conditions underscore the crucial need for improved understanding and communication in the treatment process. Furthermore, this study highlights a trend of increased burden with the duration of the animal’s therapy, particularly in dermatological diseases, emphasizing the need for tailored treatment strategies aligned with caregivers’ lifestyles. Although income did not exhibit a statistical difference in caregiver burden, a negative correlation suggests that a higher income is associated with a lower burden. Age also played a significant role, with older caregivers reporting a lower burden. Notably, cat caregivers reported a higher burden compared to dog caregivers, underscoring the necessity for further research dedicated to understanding the nuanced aspects of caregiver burden in caring for cats with various health conditions.

Future research should include conducting longitudinal studies to track caregiver burden over time, exploring differences faced by caregivers according to the stage of diagnosis and cultural characteristics, to identify cultural factors influencing caregiver burden. Educational initiatives for pet owners could also be explored to provide them with the necessary knowledge and skills to deal with the challenges of caring for pets with chronic illnesses. Recognizing the difficulties in caring for a sick pet offers an understanding of the client’s perspective, enhancing communication and potentially leading to better client adherence to the treatment plan, ultimately contributing to job satisfaction for the veterinarian.

## Figures and Tables

**Figure 1 animals-14-00276-f001:**
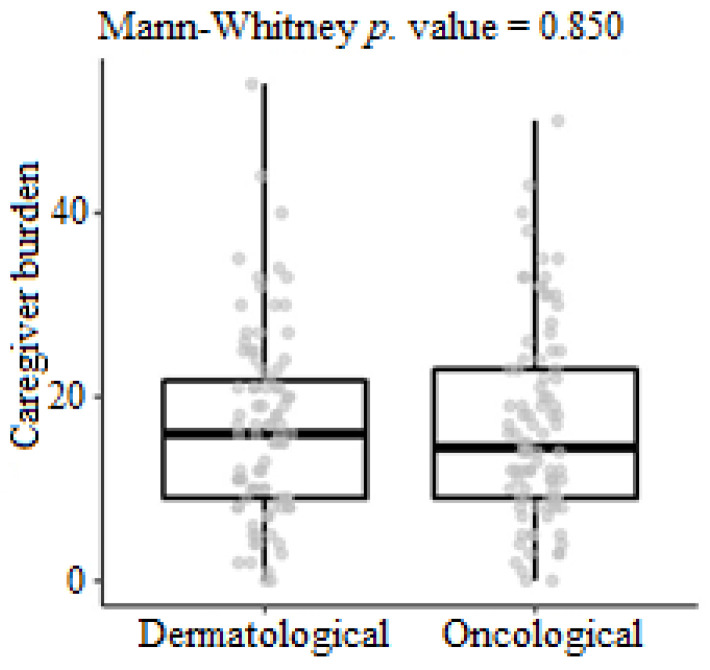
Comparison of caregiver burden levels for owners of animals with dermatological and oncological pathologies treated at HV-UFMG. Measurements of caregiver burden and boxplot showing descriptive statistics for owners of animals with dermatological and oncological pathologies, along with their *p*-value.

**Figure 2 animals-14-00276-f002:**
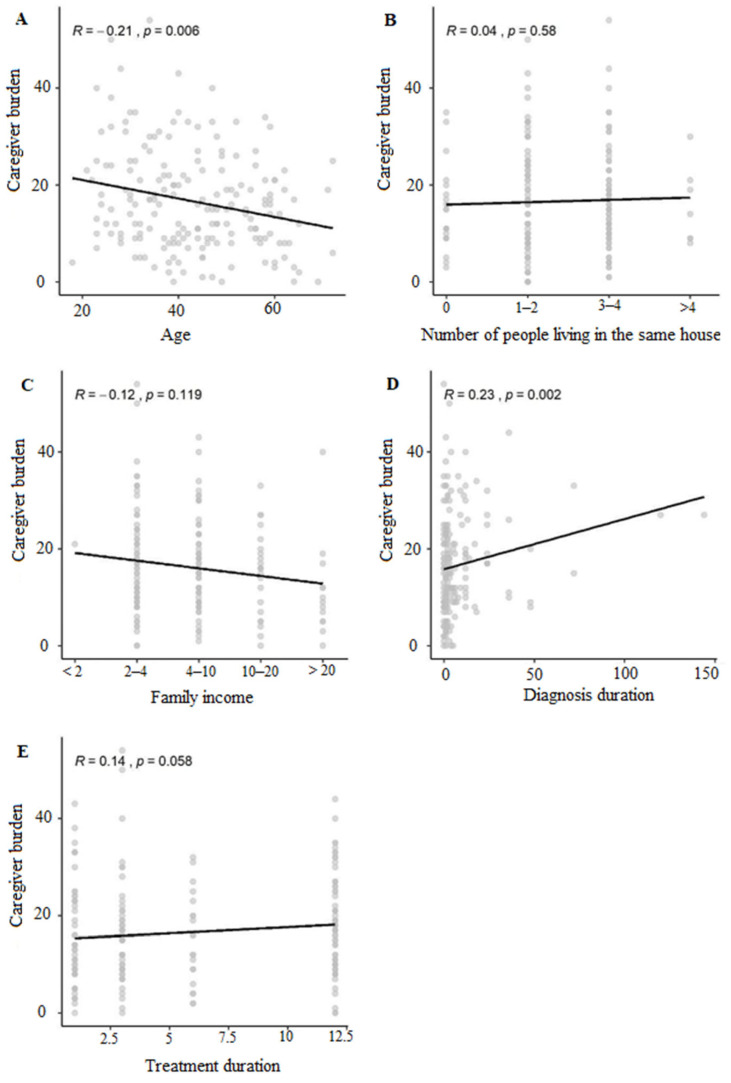
Correlation of caregiver burden with (**A**) age; (**B**) number of people living in the same house; (**C**) family income; (**D**) diagnosis duration; (**E**) treatment duration among caregivers of oncological and dermatological patients. Scatter analysis of caregiver burden correlated with age, number of people living in the same house, family income, diagnosis duration, and treatment duration, with Spearman correlation value, along with its *p*-value, for family income, following the correspondence: group 1 (up to 2 salaries), group 2 (2 to 4 salaries), group 3 (4 to 10 salaries), group 4 (10 to 20 salaries), group 5 (more than 20 salaries).

**Figure 3 animals-14-00276-f003:**
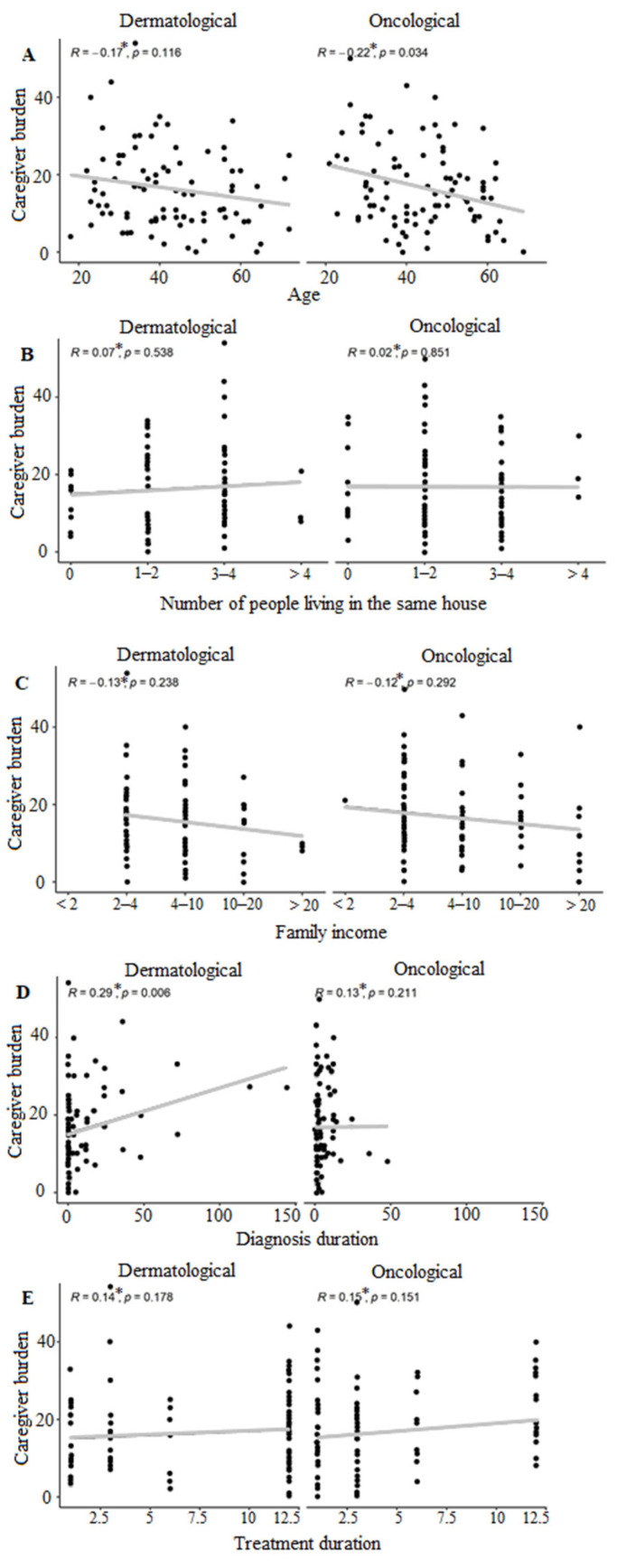
Correlation of caregiver burden of dermatological and oncological patients with (**A**) age; (**B**) number of people living in the same house; (**C**) family income; (**D**) diagnosis duration; (**E**) treatment duration. Scatter analysis of caregiver burden correlated with age, number of people living in the same house, family income, diagnosis duration, and treatment, comparing caregivers of dermatological and oncological patients, with Spearman correlation value, along with its *p*-value, for family income, following the correspondence: group 1 (up to 2 salaries), group 2 (2 to 4 salaries), group 3 (4 to 10 salaries), group 4 (10 to 20 salaries), group 5 (more than 20 salaries). * Due to the diversity of correlation values (R), these data should not be used for comparison between specialties.

**Table 1 animals-14-00276-t001:** Demographic information of the tutor and perception about the patient’s diagnosis and treatment.

Participant Identification	Responses
Age (years)	(Open-ended question)
Gender	( ) Female ( ) Male ( ) Non-binary ( ) Prefer not to disclose
Education	( ) Completed elementary school ( ) Incomplete high school ( ) Completed high school ( ) Incomplete college ( ) Completed college ( ) Postgraduate
Family Income	( ) No income ( ) Up to 2 minimum wages (R$2200.00) ( ) 2 to 4 minimum wages (R$2200.00 to R$4400.00) ( ) 4 to 10 minimum wages (R$4400.00 to R$11,000.00) ( ) 10 to 20 minimum wages (R$11,000.00 to R$22,000.00) ( ) Above 20 minimum wages (>R$22,000.00) ( ) Prefer not to disclose
Besides you, how many people live in your house	( ) 0 ( ) 1–2 ( ) 3–4 ( ) 5 or more
Are you the sole caregiver for your pet’s treatment	( ) Yes ( ) No
Pet Species	( ) Dog ( ) Cat
Diagnosis/Disease of your pet	(Open-ended question)
Do you consider the disease to be currently stabilized	( ) Yes ( ) No
How long ago was your pet diagnosed (months)	(Open-ended question)
Treatment duration so far	( ) Up to 1 month ( ) Up to 3 months ( ) Up to 6 months ( ) Up to 1 year ( ) More than 1 year

**Table 2 animals-14-00276-t002:** Modified Zarit Burden Interview.

Question	Never (0 pts)	Rarely (1 pt)	Occasionally (2 pts)	Frequently (3 pts)	Almost Always (4 pts)
1. Do you feel you don’t have enough time for yourself because of the time you spend caring for your pet?	-	-	-	-	-
2. Do you feel stressed between taking care of your pet and trying to meet other responsibilities for your family or work?	-	-	-	-	-
3. Do you feel embarrassed by your pet’s behavior?	-	-	-	-	-
4. Do you get angry when you are around your pet?	-	-	-	-	-
5. Do you feel that your pet currently negatively affects your relationship with other family members or friends?	-	-	-	-	-
6. Are you afraid of what the future holds for your pet?	-	-	-	-	-
7. Do you feel tense when you are around your pet?	-	-	-	-	-
8. Do you feel that your health has been affected because of your involvement with your pet?	-	-	-	-	-
9. Do you feel that your social life has been affected because you are taking care of your pet?	-	-	-	-	-
10. Do you feel uncomfortable having friends over because of your pet?	-	-	-	-	-
11. Do you feel that you don’t have enough money to take care of your pet in addition to the rest of your expenses?	-	-	-	-	-
12. Do you feel that you won’t be able to take care of your pet for much longer?	-	-	-	-	-
13. Do you feel that you have lost control of your life since your pet’s illness?	-	-	-	-	-
14. Would you prefer to leave the care of your pet to someone else?	-	-	-	-	-
15. Do you feel uncertain about what to do regarding your pet?	-	-	-	-	-
16. Do you think you should do more for your pet?	-	-	-	-	-
17. Do you think you could take better care of your pet?	-	-	-	-	-
18. Overall, to what extent do you feel overwhelmed in taking care of your pet?	-	-	-	-	-
Total (0–72 points) *:	-	-	-	-	-

* Score above 18 = indicative of clinically significant burden.

## Data Availability

Upon request for research purposes or data correction, the data will be made available, with the assurance that any information capable of identifying the owners of the animals will be expunged, in accordance with the guidelines of the Ethics Committee in Brazil.

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
