# Peer review of "Caregiver Burden in Small Animal Clinics: A Comparative Analysis of Dermatological and Oncological Cases"

_animals, 2024, doi:10.3390/ani14020276_

Round 1

Reviewer 1 Report

Comments and Suggestions for Authors

The article is well written, especially the literature review. The material and methods are clear. Results in the text are correct, but the diagrams induce statistical fallacy for the reader as correlation is absent. In my opinion, this is a common feature when comparing results, such as comparison of two treatment : in the vast majority, such comparisons show lack of difference. It is the case, it should not be hidden as a negative result (I exaggerate !) but presented as a result of its own. Most oncologist consider that cancer is a most serious disease. And it is actually live threatening. The present study shows that diseases threatening ability to give care, can actually be live  threatening to the animal (in that it may induce euthanasia request) and severe also for the owner. Derm is the first cause of pet owner searching a new clinic.

I see five drawbacks : 

- as opposed to the result of literature review, the questionnaire did not ask the owners what would alleviate their caregiver burden (if the authors actually asked this questions, then I did not understand their results and discussion). The unwelcome result is that authors have to make hypotheses about these reasons. Similarly, the authors did not asked wether the owners came from urban/rural areas and whether they live in a flat or had a garden. So on with important datas regarding animal welfare.

- The conclusion remains very general (also the second half of the conclusion, the two-three paragraphs need rewriting). It should be a kind of "take home message" of what is novel in the study, what could have been improved/how should be done a future study

- in the result presentation, there are two pages of 5 diagrams that are seldom commented in the text, neither in the legends. More problematic, these diagrams have a regression line on them, with R ranging from 0.02 (sic) to 0.29. It give an erroneous sensation to the reader that the results can be significantly compared. These lines should be erased. Or at least commented under each diagram that there is no correlation. The dots should be in black and the line should be in grey, not the opposite. 

- the questionnaire has to be annexed. Line 161 : comment on the modification you made, for what reasons (the subject of your study, cultural, etc.)

- Line 369 : ethical issue : you mention that data will be available on request but you had owners sign a leaflet that mentions that when finished, all the datas will be thrown away. Details are required

- the word pathology means the study of diseases; this word or related words should be changed for : disease, condition, illness, dermatosis, cancer, etc. 

Line to line 

247-248 : I don't uderstand whit this sentence adds. I read it as a "deja vu" general comment

the comment in the rest of the pages are very interrestng and relevant

Around line 272 : population size should also be commented; even though you had more responses than the initial statistical analysis required. 

283-288 : bak to the fact of these diagrams left to the reader to analyse, while the authors should proposed their analysis. 

306-308 : I don't understand the point. what do you refer to by 'in all analyses" : yours ? literature ? Similarly line 323 : what study do you refer to ? 

line 329 - 333 : I may be wrong, but I think the the author you mention findings are opposite to your. 

Line 346 : your study confirmed that prolonged caregiving to an animal has impacts on owner, other demonstrated it before. This was not the objective of you study, thus the design was not made for it, it can not be shown as a primary, but as a secondary result. Your main result and the (adequate) title of your article is on another subject, on which your study IS ORIGINAL. So, I beg you (;-), be proud of it !

- the leaflet in Brazilian should also be provided in English 

Author Response

REVIEWER 1

- Results in the text are correct, but the diagrams induce statistical fallacy for the reader as correlation is absent. In my opinion, this is a common feature when comparing results, such as comparison of two treatment : in the vast majority, such comparisons show lack of difference. It is the case, it should not be hidden as a negative result (I exaggerate !) but presented as a result of its own.

Author's response: we added information in the result section about the figures, see if it fulfills your requirement. If the reviewer prefers, we can withdraw the figures.

- as opposed to the result of literature review, the questionnaire did not ask the owners what would alleviate their caregiver burden (if the authors actually asked this questions, then I did not understand their results and discussion). The unwelcome result is that authors have to make hypotheses about these reasons. Similarly, the authors did not asked wether the owners came from urban/rural areas and whether they live in a flat or had a garden. So on with important datas regarding animal welfare.

Author's response – Unfortunately, we did not conduct these questions. These inquiries would be important to understanding certain relationships, but we hope that the data presented is sufficient for publication as it reveals interesting results.

- The conclusion remains very general (also the second half of the conclusion, the two-three paragraphs need rewriting). It should be a kind of "take home message" of what is novel in the study, what could have been improved/how should be done a future study.

Author's response – We agree and changed the conclusion, please see the new conclusion added to the manuscript.

The severity of the illness was not an influential factor in caregiver burden. However, a significant portion of caregivers experienced a clinically significant level of burden. Therefore, regardless of the specific illness the patient has, caregivers are intensely impacted when dedicating themselves to the prolonged care of their pet. Numerous efforts are invested in hoping for the recovery of a loved one, which can contribute to financial, physical, and/or psychosocial exhaustion.

This study, the first of its kind in Brazil, examined the caregiver burden among companion animal owners. The findings unveil that over a third of caregivers for animals undergoing treatment for oncological and dermatological pathologies experience a significant burden. While no statistical differences were found between these veterinary specialties, the challenges associated with chronic conditions underscore the crucial need for improved understanding and communication in the treatment process. Furthermore, the study highlights a trend of increased burden with the duration of the animal's therapy, particularly in dermatological diseases, emphasizing the need for tailored treatment strategies aligned with caregivers' lifestyles. Although income did not exhibit a statistical difference in caregiver burden, a negative correlation suggests that higher income is associated with lower burden. Age also played a significant role, with older caregivers reporting a lower burden. Notably, cat caregivers reported a higher burden compared to dog caregivers, underscoring the necessity for further research dedicated to understanding the nuanced aspects of caregiver burden in caring for cats with various health conditions.

Future research should include conducting longitudinal studies to track caregiver burden over time, exploring differences faced by caregivers according to the stage of diagnosis and cultural characteristics, to identify cultural factors influencing caregiver burden. Educational initiatives for pet owners could also be explored to provide them with the necessary knowledge and skills to deal with the challenges of caring for pets with chronic illnesses. Recognizing the difficulties in caring for a sick pet offers an understanding of the client's perspective, enhancing communication and potentially leading to better client adherence to the treatment plan, ultimately contributing to job satisfaction for the veterinarian.

- in the result presentation, there are two pages of 5 diagrams that are seldom commented in the text, neither in the legends. More problematic, these diagrams have a regression line on them, with R ranging from 0.02 (sic) to 0.29. It give an erroneous sensation to the reader that the results can be significantly compared. These lines should be erased. Or at least commented under each diagram that there is no correlation. The dots should be in black and the line should be in grey, not the opposite.

Author's response: we changed the figure 3, see if it fulfills your requirements.

- the questionnaire has to be annexed.

Author's response: we added two tables in the manuscript, please see the changes in the text. 

Line 161 : comment on the modification you made, for what reasons (the subject of your study, cultural, etc.)

Author's response: In addition to adding the questionnaire and explaining how the score was calculated, we also included this information. Please review the modifications in the text.

- Line 369 : ethical issue : you mention that data will be available on request but you had owners sign a leaflet that mentions that when finished, all the datas will be thrown away. Details are required.

Author's response – we changed the statement, see if it is better now

Upon request for research purposes or data correction, the data will be made available, with the assurance that any information capable of identifying the owners of the animals will be expunged, in accordance with the guidelines of the Ethics Committee in Brazil.

- the word pathology means the study of diseases; this word or related words should be changed for : disease, condition, illness, dermatosis, cancer, etc.

Author's response - Thank you very much for the correction, it is really not correct and we changed it to disease or condition, please see the changes in the text

Line to line

247-248 : I don't uderstand whit this sentence adds. I read it as a "deja vu" general comment the comment in the rest of the pages are very interrestng and relevant

Author's response – lines 246-248 were removed from the text – (The findings from this study hold the potential to inform future research and contribute to the development of targeted support mechanisms for pet caregivers worldwide.)

Around line 272 : population size should also be commented; even though you had more responses than the initial statistical analysis required.

Author's response: please see the changes made in the text.

Limitations of the study include the use of a convenience sample recruited via

283-288 : bak to the fact of these diagrams left to the reader to analyse, while the authors should proposed their analysis.

Author's response: please see the changes made in the text.

306-308 : I don't understand the point. what do you refer to by 'in all analyses" : yours ? literature ? Similarly line 323 : what study do you refer to ?

Author's response - In line 306-308 it was a mistake, the phrase in all analyses was removed. In line 323 we changed for: Henning et al. [16] found that the burden of care was lower in cat owners over 55 years of age and suggested that…

line 329 - 333 : I may be wrong, but I think the the author you mention findings are opposite to your.

Author's response – You are right, maybe it wasn’t clear what I meant, the authors of the article state Owners of an ill cat, examined across all illnesses represented, had greater burden (P <0.001) than the owners of a healthy cat and somewhat lower burden (P=0.013) than owners of an ill dog.. Therefore, it is the opposite of ours, this state was added, see the changes.

Spitznagel et al. [15] researched the occurrence of caregiver burden specifically in cat care-givers. It was observed that owners of sick cats had a higher burden than owners of a healthy cat, but a slightly lower burden than owners of sick dogs, results that differ from ours… Moreover, many cats do not receive routine veterinary care, even for chronic diseases, and this lack of veterinary oversight can lead to a lower demand for care for cat caregivers, but more studies to understand these differences are necessary.

Line 346 : your study confirmed that prolonged caregiving to an animal has impacts on owner, other demonstrated it before. This was not the objective of you study, thus the design was not made for it, it can not be shown as a primary, but as a secondary result. Your main result and the (adequate) title of your article is on another subject, on which your study IS ORIGINAL. So, I beg you (;-), be proud of it !

Author's response : The conclusions were changed

- the leaflet in Brazilian should also be provided in English

Author's response – We did not understand what the reviewer asked, if it the informed consent, it was provided to the editor.

Reviewer 2 Report

Comments and Suggestions for Authors

This paper looks at the burden borne by the main person responsible for caring for sick dogs or cats, in a given social context, i.e. taking into account the age and gender of the person, their financial status, or the number of people in their direct entourage). Each treatment was prescribed by a specialist in his or her field. The perceived burden of care was compared in two different patient populations within the same hospital: on the one hand, care for dermatological problems, and on the other, care for animals suffering from cancer. Cats and dogs were included, and the results were also compared between the two species.

First of all, it would be necessary to include in a table the questions actually asked to caregivers (adaptation of Zarit's ZBI grid published in 1980 with 29 items, adapted by Spitznagel for pets, but whose description specifies that 4 questions were removed from the grid initially designed for humans (1,8,11, and 14), but only 22 questions are then mentioned and it is unclear which questions were kept and which removed : "The ZBI adapted for pets thus included a total of 18 items", Spitznagel; "eg, 'Do you feel that your relative asks for more help than he/she needs?' was a question withdrawn, but corresponds to question no. 2 rather than question no. 1 in Zarit publication. Moreover, the way in which the total is calculated (point value of each answer) should be specified, since a total of more than 40 points can be reached (84 points max in Zarit study).

In addition, the study focuses on cases of either oncology or dermatology fields, on the assumption that a more severe pathology will result in a greater burden, and that a skin disease will be less 'severe' than a cancer. However, the degree of disease severity is not described in the cohort: the prognosis for these conditions (hope of recovery or not), the severity of the disease, or the intensity of the care to be given (type of treatment (surgical, chemotherapy, radiotherapy....), the number of drugs to be given, or care to be carried out at home, and the time required for this, in particular), as well as any side effects (particularly in oncology) are not specified. Furthermore, in dermatology, cancerous diseases may be diagnosed (epitheliotropic T cell lymphoma, thymoma with cutaneous repercussions, mast cell tumors, Cushing's syndrome linked to a tumour of the adrenal gland, among other examples), and it is therefore possible to have included cases of cancer in dermatology.  A table showing the diagnoses made by the specialist for the animals included, broken down into broad categories, would be necessary to provide a better understanding of the population studied (types of cancer, median prognosis for survival at the time of diagnosis, quality of life of the pet at the time the ZBI test was carried out, intensity of care or number of planned follow-up visits, cost, and for dermatology cases likewise (type of pathology, prognosis for survival at the time of diagnosis, quality of life of the pet at the time the ZBI test was carried out, intensity of care or number of planned follow-up visits, cost). And oncological cases should be withdrawn from « dermatological cases ».

Finally, it is not specified at what point in the diagnosis the questionnaire was actually completed, or whether some people completed the questionnaire several times during the follow-up. It would have been interesting to follow up this perception of burden before diagnosis and the introduction of appropriate treatment by the specialist. The fact of mixing all the results compromises the deciphering of their real significance and the relevance of the statistical studies in this study. Furthermore, this grid was validated as showing a significant burden above 18 points; however, in dermatology and cancerology (figure 1), the medians appear to be below 18, which does not seem to be a significant perceived burden. It would be interesting to further dissect the situations where the burden appeared to be greatest.

line 48: ref 1 not 2

Author Response

REVIEWER 2

- This paper looks at the burden borne by the main person responsible for caring for sick dogs or cats, in a given social context, i.e. taking into account the age and gender of the person, their financial status, or the number of people in their direct entourage). Each treatment was prescribed by a specialist in his or her field. The perceived burden of care was compared in two different patient populations within the same hospital: on the one hand, care for dermatological problems, and on the other, care for animals suffering from cancer. Cats and dogs were included, and the results were also compared between the two species.

- First of all, it would be necessary to include in a table the questions actually asked to caregivers (adaptation of Zarit's ZBI grid published in 1980 with 29 items, adapted by Spitznagel for pets, but whose description specifies that 4 questions were removed from the grid initially designed for humans (1,8,11, and 14), but only 22 questions are then mentioned and it is unclear which questions were kept and which removed : "The ZBI adapted for pets thus included a total of 18 items", Spitznagel; "eg, 'Do you feel that your relative asks for more help than he/she needs?' was a question withdrawn, but corresponds to question no. 2 rather than question no. 1 in Zarit publication.  Moreover, the way in which the total is calculated (point value of each answer) should be specified, since a total of more than 40 points can be reached (84 points max in Zarit study).

Author's response: thank you for your suggestion, please see the changes that made in the text.

- In addition, the study focuses on cases of either oncology or dermatology fields, on the assumption that a more severe pathology will result in a greater burden, and that a skin disease will be less 'severe' than a cancer. However, the degree of disease severity is not described in the cohort: the prognosis for these conditions (hope of recovery or not), the severity of the disease, or the intensity of the care to be given (type of treatment (surgical, chemotherapy, radiotherapy....), the number of drugs to be given, or care to be carried out at home, and the time required for this, in particular), as well as any side effects (particularly in oncology) are not specified. Furthermore, in dermatology, cancerous diseases may be diagnosed (epitheliotropic T cell lymphoma, thymoma with cutaneous repercussions, mast cell tumors, Cushing's syndrome linked to a tumour of the adrenal gland, among other examples), and it is therefore possible to have included cases of cancer in dermatology.  A table showing the diagnoses made by the specialist for the animals included, broken down into broad categories, would be necessary to provide a better understanding of the population studied (types of cancer, median prognosis for survival at the time of diagnosis, quality of life of the pet at the time the ZBI test was carried out, intensity of care or number of planned follow-up visits, cost, and for dermatology cases likewise (type of pathology, prognosis for survival at the time of diagnosis, quality of life of the pet at the time the ZBI test was carried out, intensity of care or number of planned follow-up visits, cost). And oncological cases should be withdrawn from « dermatological cases ».

Author's response - All assessments used in this research were based on the responses obtained from the questionnaires provided to the caregivers. The questionnaire was fully completed by the caregiver. Thus, in several cases the patient was still in the process of diagnosis, such as in the case of screening for atopy diagnosis. The intensity of patient care is subjective and varies for each caregiver (it could not be measured). There was also no planning for the number of necessary returns, as most animals would require chronic care, possibly for the rest of their lives. As an attempt to assess the caregiver's perception of the animal's illness, a question was posed about whether the caregiver considered their animal's disease stable or not. The pet's quality of life at the time of the questionnaire application was not assessed. While such an evaluation would be highly beneficial, there are already articles exploring this assessment in some previous caregiver burden studies. Additionally, the questionnaire administration took place during the hospital routine, aiming to expedite the caregiver's response time. We hope that even without these data, the article can be published and serve as inspiration for others in the field, both in Brazil and internationally, where there are currently no studies like this

Finally, it is not specified at what point in the diagnosis the questionnaire was actually completed, or whether some people completed the questionnaire several times during the follow-up. It would have been interesting to follow up this perception of burden before diagnosis and the introduction of appropriate treatment by the specialist. The fact of mixing all the results compromises the deciphering of their real significance and the relevance of the statistical studies in this study. Furthermore, this grid was validated as showing a significant burden above 18 points; however, in dermatology and cancerology (figure 1), the medians appear to be below 18, which does not seem to be a significant perceived burden. It would be interesting to further dissect the situations where the burden appeared to be greatest.

Author's response – about your concern we agree, and some future research ideas were added. Please see the changes made in the text.

line 48: ref 1 not 2

Author's response: Thank you for pointing out this mistake, the number of the reference was changed.

Round 2

Reviewer 2 Report

Comments and Suggestions for Authors

Thank you for answering most of the questions raised.

However, you did not answer the question in the materials and methods concerning the exclusion of cancer cases that may have come to dermatology consultations. Was the question in Table I (diagnosis/disease of your pet) used to separate the cases? Was the respondent's answer confirmed by the specialist?

You also did not answer the question concerning the severity of the disease for each case : yet, in the conclusion, you conclude that "The severity of the illness was not an influential factor in caregiver burden 392", which is not borne out by your results. The severity of the illness was not assessed in the study, and this sentence should be reworded accordingly.

These two points are important and deserve a response.

Finally, it would be useful in a future questionnaire to assess the complexity of treatments in progress at the time of the questionnaire (number of tablets to be taken per day, or shampoos, etc.), since you point out that "the correlation with treatment complexity remained significant, suggesting that starting with the simplest effective treatment may reduce owner strain and improve long-term management for the dog » (line 110-111).

Author Response

Question: However, you did not answer the question in the materials and methods concerning the exclusion of cancer cases that may have come to dermatology consultations. Was the question in Table I (diagnosis/disease of your pet) used to separate the cases? Was the respondent's answer confirmed by the specialist?

Answer: A criterion for exclusion was patients who had associated dermatological and oncological pathologies. They were eliminated from the study after identifying that the patient had both conditions. This perception occurred through the identification of guardians by the researchers when conducting the questionnaire and/or through the diagnosis listing that the animal had from the guardian. The veterinary specialist confirmed the responses.

Question: You also did not answer the question concerning the severity of the disease for each case: yet, in the conclusion, you conclude that "The severity of the illness was not an influential factor in caregiver burden 392", which is not borne out by your results. The severity of the illness was not assessed in the study, and this sentence should be reworded accordingly.

Answer: The sentence was removed. Since many oncological diseases involve more aggressive therapies—such as surgical procedures and chemotherapy—and have a reduced survival time compared to dermatological patients, the hypothesis is that guardians of animals facing these more severe diseases would experience a higher burden.

These two points are important and deserve a response.

Question:  Finally, it would be useful in a future questionnaire to assess the complexity of treatments in progress at the time of the questionnaire (number of tablets to be taken per day, or shampoos, etc.), since you point out that "the correlation with treatment complexity remained significant, suggesting that starting with the simplest effective treatment may reduce owner strain and improve long-term management for the dog » (line 110-111).

Answer:  Thank you for the suggestion. Indeed, this would be an important factor when assessing the impact of treatment complexity on the occurrence of caregiver burden. We will take the suggestion into account in future studies.